# Superparamagnetic Iron Oxide Nanoparticles (SPION): From Fundamentals to State-of-the-Art Innovative Applications for Cancer Therapy

**DOI:** 10.3390/pharmaceutics15010236

**Published:** 2023-01-10

**Authors:** Thomas Vangijzegem, Valentin Lecomte, Indiana Ternad, Levy Van Leuven, Robert N. Muller, Dimitri Stanicki, Sophie Laurent

**Affiliations:** 1General, Organic and Biomedical Chemistry Unit, NMR and Molecular Imaging Laboratory, University of Mons, 7000 Mons, Belgium; 2Center for Microscopy and Molecular Imaging (CMMI), Non-Ionizing Molecular Imaging Unit, 6041 Gosselies, Belgium

**Keywords:** iron oxide nanoparticles, SPION, cancer therapy, drug delivery, reactive oxygen species, magnetic hyperthermia, macrophage polarization, theranostics

## Abstract

Despite significant advances in cancer therapy over the years, its complex pathological process still represents a major health challenge when seeking effective treatment and improved healthcare. With the advent of nanotechnologies, nanomedicine-based cancer therapy has been widely explored as a promising technology able to handle the requirements of the clinical sector. Superparamagnetic iron oxide nanoparticles (SPION) have been at the forefront of nanotechnology development since the mid-1990s, thanks to their former role as contrast agents for magnetic resonance imaging. Though their use as MRI probes has been discontinued due to an unfavorable cost/benefit ratio, several innovative applications as therapeutic tools have prompted a renewal of interest. The unique characteristics of SPION, i.e., their magnetic properties enabling specific response when submitted to high frequency (magnetic hyperthermia) or low frequency (magneto-mechanical therapy) alternating magnetic field, and their ability to generate reactive oxygen species (either intrinsically or when activated using various stimuli), make them particularly adapted for cancer therapy. This review provides a comprehensive description of the fundamental aspects of SPION formulation and highlights various recent approaches regarding in vivo applications in the field of cancer therapy.

## 1. Introduction

Over the decades, cancer has become one of the major public health issues due to its global incidence. Nowadays, it is considered as the second leading cause of death in economically developed countries and was responsible for nearly 10 million deaths in 2020 (according to the World Health Organization; WHO). While conventional treatments used clinically include surgery, radiation therapy (RT), and chemotherapy, the development of nanotechnology has emerged as a powerful tool for effective diagnosis and treatment by improving drug efficacy and reducing side effects [1,2]. Through the years, subsequent developments have led to the emergence of multifunctional platforms which combine the therapeutic action from a drug or an external stimulus (magnetic or optical), with imaging capabilities such as near-infrared optical imaging (NIR), magnetic resonance imaging (MRI), or magnetic particle imaging (MPI).

Among the plethora of multifunctional nanosystems, those composed of superparamagnetic iron oxide nanoparticles (i.e., magnetite and maghemite; SPION) emerged as representative candidates with increasing uses in many biomedical fields, especially in the field of magnetic resonance imaging (MRI) where their efficiency as contrast modifiers (T_2_-weighted MRI) for magnetic resonance imaging (MRI) has been widely documented, including some clinical applications [3]. While some studies highlighted a dose-dependent gadolinium accumulation in the brain of healthy patients for the widely used gadolinium-based contrast agents [4,5,6], magnetite NPs presenting appropriate characteristics (i.e., spherical with a mean diameter below 5 nm) can be used for T_1_- or T_2_-weighted imaging with better biosafety [7,8]. Alongside MRI, the development of magnetic particle imaging (MPI) arose as a promise for tracking and quantifying SPION tracers, especially in the context of drug delivery applications or image-guided therapies [9,10].

When it comes to developing therapeutic applications, the SPION’s ability to induce local heat when submitted to an oscillating magnetic field (i.e., magnetic hyperthermia; MHT) inspired many research groups, especially in the field of oncology, because of the low tolerance of tumor cells to a moderate increase in temperature (i.e., 42–49 °C) [11]. This property can also be used to promote the release of a loaded drug upon thermal cues and can work synergistically with thermotherapy to improve the efficacy of treatment. In addition, SPION also exhibit peroxidase-like catalytic activity through the well-known Fenton reactions, which can catalyze endogenous hydrogen peroxide (H_2_O_2_) to cytotoxic hydroxyl radicals (-OH). Such chemodynamic therapy (CDT) based on the Fenton reaction has been a rapidly growing research topic in cancer treatment in recent years, some research groups evidencing possible synergies with radiation therapies. Some recent reports highlight the usefulness of SPION in magneto-mechanical therapies, in which their mechanical effects when submitted to low frequency vibrations can promote cancer cell destruction.

With regard to the promises offered by SPION in the field of oncology and the continuous development associated with this topic, the purpose of this work is to provide a comprehensive literature survey dealing with the use of magnetic nanomaterials, mainly iron oxide nanoparticles (i.e., magnetite or maghemite), as therapeutic tools for the treatment of cancer using state-of-the-art therapy.

## 2. Fundamental Characteristics of Magnetic Nanoparticles

### 2.1. Superparamagnetism

Both magnetite and maghemite are minerals from the spinel group, consisting of arrays of Fe cations and O^2−^ anions where oxygen anions are packed in a face-centered cubic (fcc) arrangement. Iron cations are located in either tetrahedral or octahedral sites (Figure 1a). Magnetite has an inverse spinel structure with both ferric (Fe^3+^) and ferrous (Fe^2+^) ions spread in the different interstitial sites. With a lattice parameter of approximately 0.84 nm [12], the inverse spinel structure is formed by 32 oxygen anions and 24 iron cations, the latter ones being distributed between 8 tetrahedral and 16 octahedral positions [13]. In stoichiometric magnetite, the Fe^2+^ to Fe^3+^ ratio is 0.5, divalent cations occupying half of the octahedral sites, while the trivalent cations are equally distributed between the tetrahedral sites and the remaining octahedral sites [14]. The general cell unit of magnetite can hence be described as ${\left({\mathrm{Fe}}^{3+}\right)}_{8}{\left[{\mathrm{Fe}}_{\frac{1}{2}}^{2+}.{\mathrm{Fe}}_{\frac{1}{2}}^{3+}\right]}_{16}O_{32}^{2-}$, where ( ) represents cations on tetrahedral sites and [ ] represents those on octahedral sites. Maghemite, usually referred as the oxidized form of magnetite, differs from magnetite by the absence of divalent cations in its crystalline structure. Trivalent cations distributed in tetrahedral and octahedral sites are accompanied by iron vacancies (◻) in octahedral sites to guarantee the global neutrality of the bulk structure. One out of six iron sites of maghemite is empty, yielding a maghemite cell unit described as ${\left({\mathrm{Fe}}^{3+}\right)}_{8}{\left[{\mathrm{Fe}}_{\frac{5}{6}}^{3+}{◻}_{\frac{1}{6}}\right]}_{16}O_{32}^{2-}$.

In their bulk state, magnetite and maghemite both exhibit a ferrimagnetic behavior, characterized by the magnetic ordering of cations in two sublattices oriented in opposite directions (Figure 1b). Such ferrimagnetic nanomaterials spontaneously divide into small regions of magnetic ordering (i.e., Weiss domains) statistically opposed to one another in order to reduce the system’s magnetostatic energy (Figure 1c). Under the effect of a magnetic field, these domains progressively align within the material until reaching saturation (magnetization saturation; M_SAT_). Once magnetized, a ferrimagnetic material retains its magnetic properties due to the non-instantaneous nature of the magnetic spins orientation with the field. This phenomenon gives rise to the remanent magnetization (M_R_), coercive field (H_C_), and hysteresis loop observed when the magnetization of a material is measured as a function of the external magnetic field [15]. Regarding magnetite and maghemite, organization in magnetic domains is usually observed for crystals with sizes above 100 nm [16].

Iron oxide nanoparticles used for biomedical applications are generally characterized by a core size varying between 5 and 60 and a hydrodynamic size below 200 nm (intravenous formulations) [17,18]. With such small size, the organization in magnetic domains is energetically unfavorable, and nanoparticles remain in a single-domain state (Figure 1d). The absence of magnetic domains results in a new magnetic behavior called superparamagnetism. Single-domain nanoparticles are uniformly magnetized and, when submitted to a magnetic field, their magnetic moments align with the field. As a consequence, superparamagnetic nanoparticles do not exhibit remanence or coercivity, and their magnetic moments disappear when the magnetic field is switched off.

The term superparamagnetism is derived from the behavior of SPION acting in the same way as paramagnetic substances, but with much larger magnitude as single-domain SPION behave like single entities with a huge magnetic moment. In the biomedical field, the superparamagnetic state is of paramount importance for two reasons: (i) superparamagnetic nanoparticles do not retain residual magnetism once the magnetic field is removed, thereby avoiding potential clogging of blood vessels due to magnetic aggregation and (ii) each single-domain nanoparticle behaves like a giant magnetic moment composed of all the individual magnetic moments of atoms which form the nanoparticle. This behavior is responsible for their high magnetic susceptibility [19].

### 2.2. Magnetic Nanoparticles Synthesis

Over the last decades, extensive research on the preparation of iron oxide nanoparticles was conducted using various approaches. Synthetic methods enabling the synthesis of iron oxide nanoparticles are usually based on a “bottom-up” approach, involving the build-up of nanoparticles from atoms, monomers, or clusters. The global formation mechanism of nanoparticles is generally described by a nucleation and growth model (Figure 2), initially described by LaMer in 1950 for the preparation of sulfur soils [20], starting from monomers generated in solution through either reduction of metallic ions or thermal decomposition of organometallic compounds [21]. Burst-nucleation occurs when the monomer concentration reaches a level (C_min_) high enough for the energy barrier to be overcome and for the nucleation phenomenon to be thermodynamically favorable. After the burst-nucleation phenomenon, the monomer concentration drops below the critical value for nucleation, ending the nucleation period. Growth of the nanoparticles (third stage) then occurs through the diffusion of monomers in solution toward the formed nuclei. This phase is governed by diffusion of the monomers and surface reaction with the nuclei.

Formation of a size distribution in solution occurs through a size focusing effect called Ostwald ripening, causing the dissolution of nanoparticles with size below the critical radius, hence regenerating monomers which can redeposit onto the surface of bigger nanoparticles [22].

In recent years, numerous synthetic routes have been established to obtain iron oxide nanoparticles (primarily magnetite or maghemite) with various structural features. Among these routes, coprecipitation and thermal decomposition are the most widely used ones, especially for the preparation of nanometric scale (<20 nm) iron oxides. Other methods, such as micro-emulsion, sol–gel and polyol synthesis, and hydrothermal synthesis, with varying degrees of effectiveness, have also been developed [23].

Iron salts coprecipitation in alkaline media is generally considered the most efficient way to produce nanoparticles readily dispersible in water, mainly due to its ease of implementation and large volume capacity [24]. Control over particle size distributions and magnetic properties is obtained by performing the reaction in the presence of capping agents such as polymers (dextran, polyacrylic acid, diethylene glycol) and by adjusting experimental conditions (stoichiometry, pH, ionic strength, temperature, etc.) [25]. Thermal decomposition, however, is often chosen over coprecipitation when greater control over nanoparticle size and morphology is desired. The decomposition of iron organometallic precursors at elevated temperature in the presence of organic surfactants (i.e., oleylamine, oleic acid, etc.) affords the obtention of narrow size distributions as well as highly crystalline nanoparticles. Moreover, the main advantage of the method is its ability to yield particles with exotic morphologies (Figure 3).

**Figure 3 pharmaceutics-15-00236-f003:**
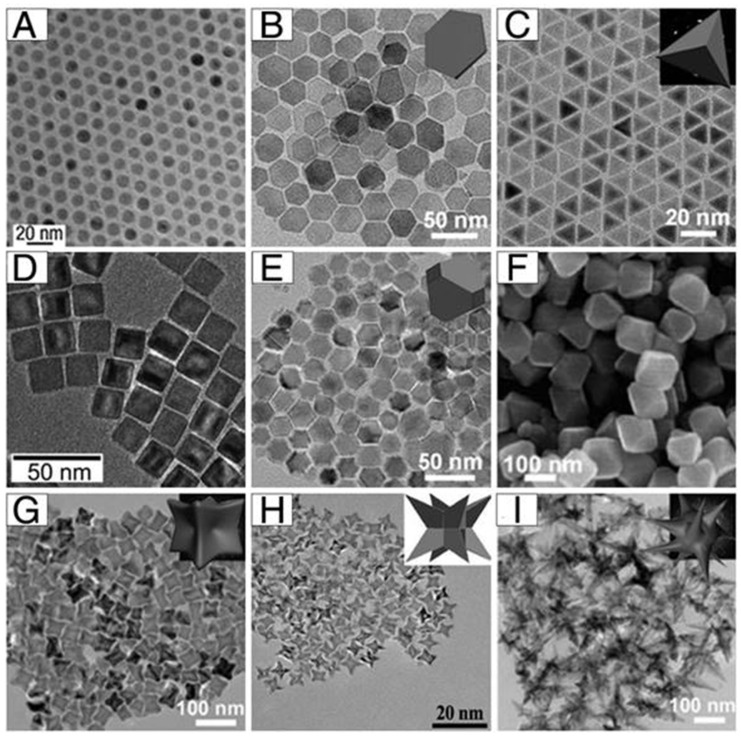
Examples of particle morphologies: (**A**) nanospheres; (**B**) plates; (**C**) tetrahedrons; (**D**) cubes; (**E**) truncated octahedrons; (**F**) octahedrons; (**G**) concaves; (**H**) octapods; (**I**) multibranches. Reprinted from [26].

Although synthetic procedures occurring in batch conditions remain the most prevalent methods used for the synthesis of nanomaterials, their use for the production of large volumes is still limited due to inter-batch variations and high equipment costs for large-scale manufacturing [27]. Instead of scaling-up the currently existing classical reactors from the laboratory to the industrial scale, continuous production using micro- or millimetric reactors has recently emerged as a reliable technology for the production of SPION [28,29]. The major advantages of continuous flow chemistry include strict control of reaction parameters, improved heat transfer due to reduced reactor dimensions, and the ability to operate continuously at steady state [30,31]. Both coprecipitation and thermal decomposition methods have been adapted using continuous flow conditions to produce various types of nanoparticles [32,33,34,35].

### 2.3. Particle Dimensions

Particle size is the main physico-chemical property that can be tuned to reach optimal performance regarding applications in the field of cancer therapy as it directly affects other properties, such as magnetism, surface area, and particle composition. For a given shape, variations in the magnetic properties of particles as a function of size have been demonstrated to be the combined result of size and composition. Salazar et al. described a core-shell model, i.e., a magnetite core covered by a maghemite shell for particles larger than 20 nm, while the smallest (<20 nm) are described as nonstoichiometric magnetite nanoparticles since the fractional volume of magnetite decreases with the particle size [36]. Moreover, the presence of a spin-canted shell (magnetically “dead” region) onto the surface of very small iron oxide nanoparticles (<5 nm; in which surface atoms represent 50% of the particle volume) is another parameter known to be responsible for their reduced magnetization [7].

Besides its influence on physico-chemical properties, the size of nanoparticles has a major impact on their biodistribution upon administration through intravenous injection. Small-sized nanoparticles (i.e., with hydrodynamic diameter below 10 nm) tend to be quickly excreted from blood circulation through renal clearance. Increasing particle size (i.e., using particles in the 10–100 nm size range) is generally considered to be the optimal range due to two criteria: (i) particles circulate for a longer period within the bloodstream, and (ii) they can easily penetrate through larger fenestrations of the blood vessels, hence preferentially accumulating in target tissues such as tumors [19,37]. The latter, generally known as the enhanced permeability and retention (EPR) effect, is a heterogeneous phenomenon occurring to various extents in different kinds of solid tumors, characterized by highly permeable vasculature promoting the enhanced permeability of particles, as well as impaired lymphatic drainage promoting their enhanced retention [38,39,40].

Similarly to particle size, the shape of nanoparticles is another physical parameter directly dictating their pharmacokinetic and biodistribution profile, as well as their toxicity. Moreover, cellular interaction (cellular uptake, uptake kinetics, and mechanism) is also governed by particle shape [41]. In recent years, the increasing control over nanoparticle synthetic routes made it possible to achieve the preparation of anisotropic iron oxide nanoparticles with different morphologies such as cubes, rods, disks, flowers, tetrapods, etc. [26,42,43] (Figure 3). Although spherical nanoparticles remain the dominant shape due to ease of synthesis, more exotic architectures such as needles or disks have been demonstrated to exhibit favorable properties in the field of cancer nanomedicine [44]. Worm-like or filamentous nanoparticles, for instance, have been demonstrated to circulate continuously in mice or rats for one week, thereby extending the time window for potential interaction with tumors [45]. Elongated particles generally exhibit reduced cell uptake due to reduced formation of the actin cup which is an important prerequisite for phagocytosis by macrophages [46]. As an example, Naumenko et al. compared the biodistribution profiles of nanocubes, nanoclusters, and nanorods coated with F-127 poloxamer [47]. Slower uptake of nanorods was observed in the liver and the spleen, compared to the two other systems, and was linked with lower sequestration by macrophages.

### 2.4. Coating

Bare nanoparticles are characterized by high surface energy and tend to naturally aggregate due to Van der Walls forces and magnetic dipolar interactions, described by the Derjaguin–Landau–Verwey–Overbeek (DLVO) theory of colloid stability [48]. Surface modification with an appropriate coating is hence crucial to guarantee their stability in physiological and biological fluids. 

Typical capping agents used for the stabilization of SPION are generally composed of:An anchoring group, i.e., a moiety having a good binding affinity toward the nanoparticle surface. These groups (Figure 4a) include, for example, carboxylates, dopamine, phosphonates (mono- or bidentate), 2,3-dihydroxybenzamide, hydroxamate, siloxane, etc.A hydrophilic moiety exhibiting either steric or ionization characteristics necessary to stabilize the nanoparticles. Such (in)organic capping agents can be attached onto the nanoparticle surface either by physisorption (Figure 4b) or by covalent binding through the anchoring groups (Figure 4c).

The list of commonly used electrostatic stabilizers includes citric acid, ammonium salts (cetyltrimethylammonium bromide (CTAB), tetramethylammonium (TMAOH)), sodium salt (sodium dodecyl sulfate (SDS)) or amino acids (L-lysine, aspartic acid [49,50], etc.).

With regard to biomedical applications, stabilization of SPION through electrostatic repulsion remains less effective than steric stabilization due to increased ionic strength encountered in a biological medium, leading to destabilization of colloidal suspensions. Polymers are among the most widely used coating agents to provide nanoparticles with colloidal stability through steric stabilization [51]. Synthetic polymers, such as polyethylene glycol (PEG), polyvinyl alcohol (PVA), polyvinyl pyrrolidone (PVP), etc., are particularly efficient for the stabilization of nanoparticles. The nature of the polymer is directly responsible for the nanoparticle interaction with the biological media, therefore strongly impacting the biodistribution kinetics and fate of SPION for clinical applications.

In the field of cancer therapy, particle PEGylation is generally fostered given the ability of PEG to confer “stealth” to nanomaterials. PEGylated nanomaterials are characterized by longer half-lives due to the hydrated cloud formed by the polymeric backbone onto their surface, resulting in objects with higher hydrodynamic volume, hence less likely to be filtered by kidney clearance [52,53]. PEG coating also shields nanomaterials against recognition by blood proteins (i.e., opsonins), preventing their adsorption and phagocytosis through the mononuclear phagocyte system (MPS) [54]. Consequently, a significantly higher therapeutic outcome is expected from long circulating nanomaterials due to their increased likelihood of interacting with receptors on cell surfaces (in the case of vectorized nanoparticles) or accumulating within tumors through the EPR effect. The molecular weight of PEG and its grafting density onto the particle surface are additional parameters impacting the uptake of nanoparticles by cells as well as their pharmacokinetics upon intravenous injection. Image-guided follow-up of SPION biodistribution is easily achievable, using magnetic resonance imaging, thanks to their strong efficacy as contrast enhancers [3]. Several reports evaluated the biodistribution of PEGylated iron oxide nanoparticles. Xue et al. demonstrated that the PEG layer has a key role in determining the half-life of 14 and 22 nm spherical SPION [55]. This study showed an increased half-life time for 14 nm SPION coated with PEG_5kDa_ compared to the same SPION coated with short PEG chains (i.e., PEG_2kDa_), while increasing the core size (PEG_5kDa_-coated 20 nm SPION) had a less significant influence on circulation time. In another study from Lazaro-Carrilo et al., 15 nm SPION modified with 2 different molecular chain lengths (5 kDa and 20 kDa) were evaluated as long-lasting MRI probes for cancer diagnosis [56]. While in vitro experiments showed no cytotoxicity on multiple cell lines, as well as similar cellular uptake for both formulations, in vivo experiments demonstrated a higher safety for PEG_5kDa_-coated SPION. In vivo MRI results on mice tumor xenograft models have shown longer retention times within the tumor for this formulation, with respect to ferumoxytol. Sub-5 nm SPION coated with PEG of different chain lengths (i.e., PEG_800_, PEG_2kDa_, and PEG_5kDa_) also exhibited an increased circulation time when using longer PEG chains (demonstrated by T_1_-weighted MRI and fluorescent/optoacoustic imaging) [57].

### 2.5. Protein Corona

Another factor involved in the biological fate of nanosystems when administered to living systems is the formation of the so-called protein corona (PC) onto their surface. This corona corresponds to the shell of biomolecules (including proteins, enzymes, lipids, etc.) wrapping nanoparticles upon their entering within biological fluids [58]. As a result, a distinction should be made between the nanoparticle formulation designed by scientists and what is exactly recognized by cells in the bloodstream, i.e., the PC which will dictate the nanoparticle–cell interactions [59]. Depending on the binding strength between biomolecules and the system’s surface, PC is generally described as a dynamic multi-layered structure (Figure 5) [60].

**Figure 5 pharmaceutics-15-00236-f005:**
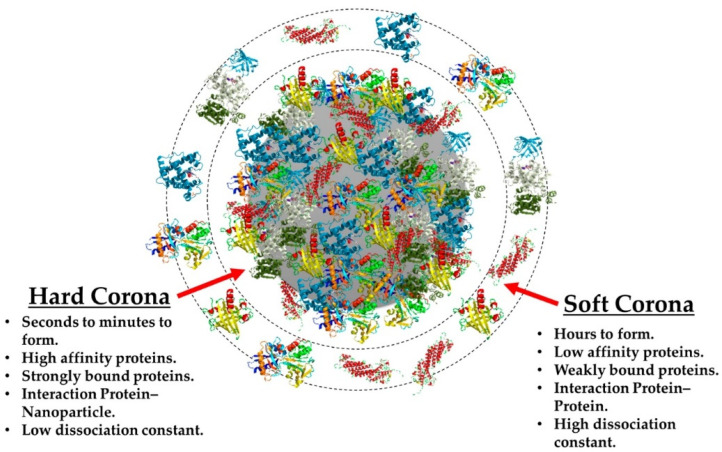
Schematic comparison of the main differences between the “hard” corona and the “soft” corona. Reprinted from [60].

Formation of a “hard” corona consisting of high-affinity proteins (i.e., apolipoproteins, fibrinogen, and albumin) irreversibly bound with the particle surface is rapidly achieved within minutes upon incubation in a physiological fluid [61]. Weak protein–protein interactions occur between proteins from the hard corona and low-affinity proteins in the second layer, generally designed as the “soft” corona. Both layers are in a dynamic process involving continuous adsorption/desorption of proteins on and off the particle with varying kinetics depending on the contact frequency between the protein and nanoparticle as well as particle characteristics, such as surface area, shape, radius of curvature, etc. [62]. Depending on the type of nanomaterial, it has been demonstrated that the corona can have a size ranging between 3 and 15 nm [61]. Therefore, understanding the formation of PC around any nanomaterial is imperative to the applications of SPION as therapeutic or diagnostic nanomedicines as it may have advantages or disadvantages regarding their efficacy. It has been found that the entry route (i.e., inhalation, intravenous injection, oral administration) of nanosystems directly influences the composition of the PC and its evolution overtime in vivo [63,64]. In addition, it has been shown that loss of targeting properties, as well as a decrease of drug release (i.e., release of camptothecin from SPION or paclitaxel from albumin) can be caused by the formation of PC, leading in both cases to a rather general decreased outcome regarding the therapeutic efficacy [65]. Ultimately, interactions at the cellular level are defined by the whole biological entity formed by the combination of PC with the nanoparticle, hence affecting their toxicity, immune response, uptake by cells, and biodegradation [58,59,66]. 

## 3. Cancer Therapy

A substantial number of studies on the use of SPION for various types of cancer-related therapies using various cell death mechanisms were proposed. Current chemotherapeutic drugs suffer from a lack of specificity, indiscriminately causing damage to both healthy cells and cancerous cells. Associating drugs with nanocarriers such as iron oxide nanoparticles has proven to be an efficient method to overcome the drawbacks of chemotherapy, while providing a way to follow the drug release by MRI as well as to trigger the release thanks to various stimuli. Iron oxide nanoparticles can generate mechanical forces or heat upon exposure to an alternating magnetic field (AMF) at either low frequency or high frequency, respectively, thereby inducing destructive effects on tumor tissues. Another characteristic feature of SPION which is consistently gaining attention is their ability to produce reactive oxygen species (ROS) through the iron-catalyzed Fenton and Haber–Weiss reactions. 

In a biological medium, the Haber–Weiss’ reaction is considered thermodynamically unfavorable in the absence of any catalyst but can generate highly reactive radicals (OH^•^) by the iron-catalyzed Fenton reaction. Indeed, a reaction between hydrogen peroxide (H_2_O_2_) and iron (II) (Fe^2+^) leads to the formation of hydroxide (OH^−^) and hydroxyl radicals (OH^•^) in the Fenton’s reaction (Equation (2)), whereas the Haber–Weiss reaction leads to the generation of hydroxyl radicals and hydroxide from an iron-catalyzed reaction between H_2_O_2_ and the superoxide ion (Equations (1)–(3)) [67]:Fe^3+^ + O_2_^−^ → Fe^2+^ + O_2_(1)
Fe^2+^ + H_2_O_2_ → Fe^3+^ + OH^−^ + OH^•^(2)
O_2_^−^ + H_2_O_2_ → OH^−^ + OH^•^ + O_2_(3)

Such ROS are well-known for their tremendous importance in the regulation of the tumor micro-environment (TME). Production of ROS induced by cell exposure to iron oxide nanoparticles, either alone or catalyzed using light (UV-visible or infrared light) or ionizing radiation (X-rays, protons, etc.), is a promising approach which can be used for cancer treatment. 

### 3.1. Drug Delivery

Nowadays, the control of drug delivery to localized targets has become a major focus of biomedical research, especially in the field of cancer therapy [68,69]. Many nanosystems have been proposed as nanocarriers, and magnetic nanoparticles (i.e., magnetite and maghemite; SPION) appear as one of the most promising thanks to their biocompatibility and (super)paramagnetic properties, such as magnetic properties allowing for the combination between imaging modalities and specific release of drugs using either local (i.e., pH, conjugation of biomarkers, …) or external stimuli (i.e., external magnetic field).

The use of an image-guided drug delivery system is expected to provide a visualization of the spatial distribution of drugs and monitor its effective accumulation within tissues, possibly allowing for real-time adjustments of delivered doses in order to remain within the therapeutic window. The benefits of these non-invasive strategies are numerous and include improved treatment efficacy and reduced side effects. Most of the studies proposed the use of MRI [70,71,72,73] as a monitoring technique, and some recent reports mention Magnetic Particle Imaging (MPI) as a promising alternative. As SPION tracers are exogenous, this imaging technique has nearly no background in comparison with contrast-modified MRI [74,75]. The signal arising from SPION can be processed to reflect their spatial location and concentration, opening the doors for quantitative imaging with good spatial resolution (≈1 mm) and sensitivity (≈100 µM Fe) [10,76]. The use of MPI was proposed by Jung who developed olaparib-loaded exosomes as a platform for drug delivery to hypoxic cancer cells [77]. In another study, Liang developed apoptosis-specific tracer (Alexa Fluor 647-AnnexinV tagged SPION) and used MPI to accurately detect and quantify apoptotic tumor cells [78]. Using an MPI scanner, they could evidence a proportionality between the imaging signal and the number of apoptotic cells. Zhu [79] designed DOX-loaded SPION/poly(lactide-co-glycolide acid) (PLGA) core-shell nanocomposite as a drug delivery system. In their work, a good correlation between the nanoparticle-induced change in MPI signal and the release rate of DOX over time upon acidolysis was found, establishing quantitative monitoring of the release process in cell culture.

Passive targeting of solid tumors with nanoparticles is based on the assumed vascular permeability of the tumor microenvironment (i.e., by EPR effect) [39,40,80,81]. However, some reports highlight an inefficient process with tumoral accumulation of around one percent of the injected dose, even for xenograft tumors [61]. To improve the uptake and/or the accumulation, active targeting strategies have been explored through the attachment of antibodies, aptamers, or peptides to the nanocarrier surface. Such tumor-targeted strategies are expected to have limited impact on healthy tissues, resulting in higher therapeutic efficacy and lower toxicity. Nosrati provided a pyoverdine-based SPION and bound them to MCU1 aptamer as a DOX-delivery system [82]. In vivo studies highlighted significant tumor inhibition and survival rate in the mice receiving a single-dose aptamer-SPION in comparison with the non-targeted particles. In another study, Jia et al. prepared glioma-targeted vesicles by incorporating SPION and curcumin (Cur) into exosomes modified with neuropilin-1-targeted peptide (RGERPPR, RGE) [83]. Along with strong loading capacities, these natural vesicles [84] are able to cross the blood–brain barrier (BBB). When administered to orthotopic glioma models, they found promising results for targeted imaging and therapy of glioma, highlighting a potent synergistic antitumor effect between MHT and cur-mediated therapy. In their work, de Lavera et al. noted a synergetic effect in vitro when combining both the ligand neuregulin and a bispecific antibody fragment to SPION [85]. They proposed their system as a good alternative for administering camptothecin. Recently, specific drug delivery using the membrane of cancer cells, specifically that of homologous tumors, has emerged as a promising targeting method [72]. This bio-inspired strategy showed great potential for targeting specific tumors by adjusting the corresponding type of modified cell membrane. Zhu proposed a theranostic platform based on SPION covered with cracked cell membrane derived from a specific tumor type [86]. A combination with DOX showed strong potency for tumor treatment in vivo. A similar strategy was proposed by Meng for the specific delivery of peptides (namely PD-L1 inhibitory peptide and MMP2 substrate peptide) [87]. They showed superior targeting capacities with TPP1 peptide being delivered and released to the tumor microenvironment through the homotypic effect of tumor cell membrane and specific digestion by the tumor-specific enzyme MMP2.

It is believed that the percentage of drug release in a specific location can be increased by using stimuli-responsive systems [88,89,90]. A common strategy is to take advantage of the pH difference between tumoral (slightly acidic) and healthy tissues. As a result, strategies implying the use of pH-sensitive bonds (e.g., hydrazone) or pH-sensitive coatings (e.g., polymers) were developed [91]. An interesting releasing approach was proposed by Liu and collaborators who developed mesoporous silica-coated SPION with programmable DNA hairpin sensor «gates» as tumor-cell-specific drug delivery systems [92]. Drug release was promoted by endogenous miRNA-21 overexpressed in tumor cells (HepG2, human liver tumor cells), which served as a specific key to unlock the system through hybridization with programmable DNA hairpin. Cai developed a reduction and pH dual-sensitive targeted (anti-glypican-3 antibody) polymeric micelles incorporating sorafenib and SPION for the treatment of hepatocellular carcinoma [93]. The efficiency of the nanocarrier was shown in vivo to display prominent anticancer effects and a low systemic side effect in an animal study (mice baring subcutaneous HepG2 xenograft). To reduce the side effect of paclitaxel and enhance its accumulation at the tumoral site, Ding prepared a novel redox-responsive magnetic nanocarrier based on disulfide-linked amphiphilic polymer [94]. The nanocarrier significantly improved intracellular uptake and enhanced tumoral accumulation, with an effective inhibition of tumor cells in vitro and in vivo through sensitive redox response, resulting in effective PTX release.

Taking advantage of the magnetic properties of iron oxide nanoparticles, some groups proposed site-specific drug delivery by inducing SPION accumulation by the action of a localized external magnetic field. In this context, SPION can be used either as individual nanoparticles (i.e., monocore systems [95]; in that case the magnetic response of SPION depends on their physicochemical properties, more specifically, their saturation magnetization which must be as high as possible) or as magnetic nanoassemblies encapsulated in macromolecular matrices (i.e., multicore systems) [96]. SPION organization within stabilized nano/micro-assemblies appears as a promising strategy to reach efficient response toward an external magnetic stimulus. For example, Qi et al. recently designed a reticulocyte (RTC) exosomes-based SPION clustering strategy for tumor-targeted drug delivery [84]. The tumoral magnetic-targeting ability was shown in vivo using Cy5.5-labelled SPION-exosomes through a significant increase of the fluorescence signal at the tumoral area when an external magnetic field (1 T) was applied. Micro/nano-scale robots [97] can effectively convert magnetic energy into locomotion [98] and can be used to support drug delivery vehicles, particularly in the case of cancer [37]. Recent progress of untethered mobile micromotors highlighted huge potential for targeted drug delivery in vivo. In the particular case of cancer treatment, Kim and co-workers proposed a hydrogel sheet-type magnetic retrieval intraocular microrobot for the treatment of retinoblastoma [99]. The developed system was made of a therapeutic layer of gelatin/PVA composed of PLGA–DOX drug particles and a SPION layer composed of PEGDA containing SPION. The potential for the vitreous migration of the microrobot and the therapeutic effect against retinoblastoma Y79 cancer cells was shown using ex vivo bovine vitreous and in vitro cell tests.

### 3.2. Magnetically Activated Therapy

In recent years, it has been demonstrated that SPION’s magnetic properties can be modulated through their shape and composition, so that they can be mechanically actuated, attracted to a region of interest, or used to generate a local heating. This is of particular interest in the context of mechanobiology, where SPION can be used to apply local forces on biological specimens using extremely low frequency magnetic field (ELMF) and to study the cellular response of tumor cells. Destroying cancerous cells or tumors through magneto-mechanical effect is a growing field of research. The desired effect is reached by means of low frequency mechanical vibrations of SPION (typically highly anisotropic particles), which induce a very localized effect with potentially few side effects. Compared to hyperthermia-based methods, the magneto-mechanical treatment mode requires lower frequencies (i.e., ELMF between 2 and 20 Hz compared to kHz frequencies regarding MHT), which could appear relevant from a safety point of view. The details of this particular technique are out of the scope of this review, but [100] provides more information. 

Li and collaborators investigated the effects of SPION on tumor cells under extremely low frequency magnetic field exposure conditions (ELMF; 2–20 Hz at intensities varying between 0.1–20 mT) [101]. They found significant morphological changes in tumor cells treated with SPION in combination with ELMF; destructive effects of SPION and ELMF on tumor tissues were further determined by the pathophysiological changes observed in vivo. Maximal effect was observed at the highest frequency (20 Hz) as well as the highest field strength (20 mT). More recently, Lopez et al. found that CCK2-targeting 6 nm-SPION (modified with gastrin peptide) can disrupt the tumor microenvironment through mechanical forces upon low frequency rotating magnetic field exposure [102]. They assumed local mechanical damage of lysosome membranes as a consequence of a torque produced by the SPION inside lysosomes under a rotating magnetic field. Martínez-Banderas et al. promoted cancer cell death by combining the chemotherapeutic effect of DOX-modified iron nanowires with the mechanical disturbance under a low frequency AMF [103]. In their study, they found a synergistic cytotoxic effect arising from the combination of a chemotoxic and magneto–mechanical treatment (1 mT, 10 Hz). Guo found magnetic hybridized microparticles (i.e., Fe_3_O_4_/BSA/rSiO_2_; Figure 6) with a flagellum-like surface as force-mediated cancer treatment [104]. The cell killing efficiency increased with increasing magnetic field frequency (≤3 Hz), vibrating magnetic field (VMF) exposure time (≤1.5 h), and microsphere concentration. In combination with VMF, their system efficiently inhibited mouse tumor growth when compared to microspheres without a VMF.

**Figure 6 pharmaceutics-15-00236-f006:**
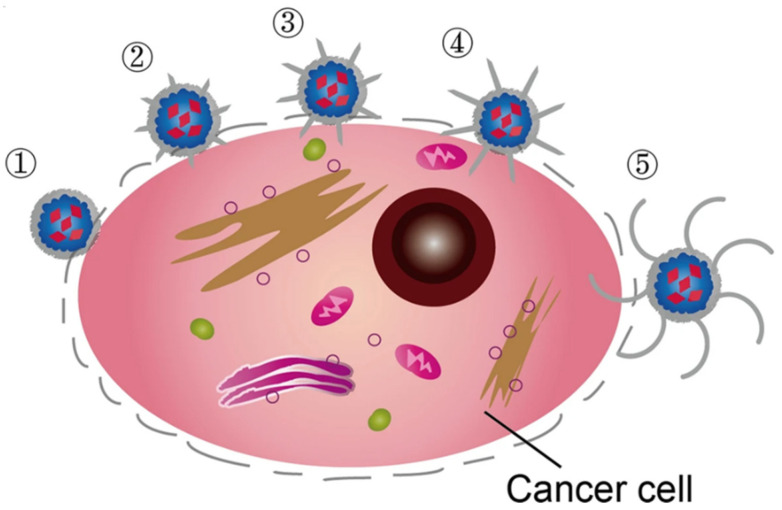
Schematic illustration of the abilities of magnetic microspheres with different sharpness to pierce the cancer cell membrane. Fe_3_O_4_/BSA, Fe_3_O_4_/BSA/SrSiO_2_, Fe_3_O_4_/BSA/MrSiO_2_, Fe_3_O_4_/BSA/LrSiO_2_, and Fe_3_O_4_/BSA/CrSiO_2_ are numbered ➀, ➁, ➂, ➃, and ➄, respectively. Reprinted from [104].

Magnetic hyperthermia is an extensively studied form of hyperthermia therapy that involves the excitation of a magnetic substance through an AMF-induced movement. Consequent to such exposure, heat can be released in the vicinity of the SPION. In the context of cancer therapy, this modality of treatment implies locally increasing the temperature to a range between 39 and 45 °C inside the tumor zone in order to kill cancerous cells or to sensitize them against other modalities of treatments. The major advantage of such a technique in comparison of other common modalities of treatment for cancer (e.g., surgery, chemotherapy, radiotherapy) is an increased specificity in the localization of the deposited effect. In other words, MH implies that solely the area that has internalized a magnetic substance and that is targeted with an AMF would be exposed to an increase in temperature and consequently to a biological effect (assuming that the Hergt’s biological safety limit is not exceeded which is likely to be true in a clinical setup [105]).

Several anticancer mechanisms of actions have been described for hyperthermia through the years including perturbation of the DNA repair process [106], fluidification of the cellular membranes [107], induction of programmed cell death [108,109,110,111], denaturation of proteins [112], modulation of the expression and activity of certain proteins [113], and modulation of the TME characteristics [113,114]. It must be mentioned that despite numerous investigations about the anticancer potential of SPION as an MH platform, several issues remain critical for their use in the context of cancer treatment by MH. These issues include: 

The optimization of the physico-chemical parameters of the SPION platforms (incl. material, magnetic behavior when exposed to a magnetic field, size, shape, and concentration) and the AMF parameters (incl. amplitude and frequency, the time of exposure to the AMF) with regards to the biological effectiveness of MH on cancerous cells [115].

The obtention of a high biocompatibility for the nanoplatform when not exposed to an AMF with regards to the biological environment.The preparation of a highly tumor-specific accumulating platform.

With respect to the abovementioned critical issues, an illustrative study can be seen in the work of Kandasamy and colleagues which focused on the heat-generating capability of 9 nm core sized SPION functionalized with different combinations of trimesic acid (TMA), pyromellitic acid (PMA), 2-aminoterephthalic acid (ATA), and terephthalic acid (TA) coatings under various concentrations (from 0.5 to 8 mg/mL) and AMF parameters (from approximately 100 to 1000 kHz). This systematic study highlighted the complexity of the as-mentioned parameter interactions in a context of optimization of the heating capability of the SPION. As an illustration of this complexity, the specific absorption rate (SAR, which relates to the amount of thermic energy release per quantity of SPION) of TA-ATA bifunctionalized SPION have displayed mixed decrement/increment patterns for various concentrations in considered magnetic fields. 

Another strategy for the optimization of a SPION platform regarding their heat-generating ability is the preparation of mixed ferrites having higher magnetization values, thanks to the presence of other ions such as manganese, cobalt, etc., within their oxide core [116,117]. As an example, this strategy was described by Kowalik et al. who took advantage of the doping of Fe_3_O_4_ NPs with yttrium trivalent ions [118]. By using this strategy, it was demonstrated that Fe_3_O_4_ NPs doped with 0.1% Y^3+^ ions dramatically decreased the viability of 4T1 cancer cells compared to Fe_3_O_4_ without doping. When the cells were subjected to the AMF for 30 min, using concentrations of 35 µg/mL and 100 µg/mL of the Fe_3_O_4_:0.1% Y NPs, the authors found a stronger decrease of cell viability for Y-doped NPs in comparison with pure Fe_3_O_4_ NPs. Another SPION doping strategy was presented in the paper of Phong et al. [119]. In this study, cobalt (Co) doped magnetite (Co_x_Fe_3−x_O_4_) NPs were synthesized and evaluated. Even if this study did not evaluate the therapeutic potential of this platform on biological material, the heat induction capability of this platform made it a suitable candidate for MH.

Even if the specificity of the action of SPION-induced MH on cancerous cells is passively ensured by a reduced heat resistance capacity in comparison with healthy cells [111,117], the specificity of the action of MH should be increased to minimize the damage to sensitive healthy areas. One of the strategies available to reach this goal is to properly functionalize the SPION with active targeting elements, such as peptides, nanobodies, or antibodies. Such a strategy was evaluated by Legge [120]. In this paper, the production of SPION with a biocompatible silica coating and antibody conjugation to target integrin αvβ6, a well-studied oral squamous cell cancer biomarker, was undertaken. The MH potential of this platform was also studied by exposing it to an AMF, which led to a reduced cell survival rate on cell line overexpressing αvβ6 in comparison with a similar cell line that did not overexpress αvβ6. 

The above-cited studies considered the sole role of MH for the treatment of cancer. However, it should be noted that it has been recently proposed to use MH to induce a cytotoxic effect on cancerous cells and, at the same time, sensitize them to other therapeutic strategies such as chemotherapy. Owing to the ability of SPION to produce heat following exposure to an external alternative magnetic field (AMF), one may expect improved drug release rates following thermal activation. To control the dosage of the cargo released by an AMF actuation, Chen proposed a superparamagnetic doped iron oxide core@shell nanoparticle (MnFe_2_O_4_@CoFe_2_O_4_) embedded within a mesoporous silica shell [121] and modified with a thermo-responsive molecular gatekeeper containing aliphatic azo group (i.e., thermal-labile 4,4′-azobis(4-cyanovaleric acid moiety) to regulate the cargo release. Efficient stepwise cargo release was achieved by applying multiple sequential AMF. On the same basis, El Boubbou et al. recently developed a PVP-coated SPION platform loaded with doxorubicin [122]. The doxorubicin-loaded and PVP-coated SPION were shown to be much more potent to the cells than the doxorubicin-free PVP-coated SPION platform when this platform was evaluated in vitro on MDA-MB-23 cancer cells under AMF. Even in brief experiment periods (15 min) at relatively modest dosages (0.5 mg·mL^−1^) and clinically relevant magnetic fields (H0 = 170 Oe and frequency = 332.8 kHz), the loss in cell viability was noticeable.

Another doxorubicin-loaded SPION platform was proposed by Sabouri et al. [123]. This investigation was about the dual osteosarcoma treatment-bone regeneration potential of a SPION platform encapsulated in a biogel. The resultant particles were bioactive as a result of the development of a hydroxyapatite layer on their surface while doxorubicin cargo was used to produce anticancer effects. This platform was tested in vitro on a human osteosarcoma cell line (MG63) which showed that at pH 5.1, doxorubicin-loaded particles released their content in a regulated manner, which was accompanied by a substantial loss in cell viability.

### 3.3. ROS-Mediated Therapies

#### 3.3.1. Macrophage Polarization

Among the many applications of magnetic nanoparticles in cancer therapy, recent studies have found biological effects arising from the intrinsic properties of the iron oxide core being able to induce the generation of reactive oxygen species (ROS) and to modulate intracellular redox mechanisms or iron metabolism [124]. Tumor-associated macrophages (TAMs) are key components of tumor metastasis, playing a significant role as metastasis promoters in the TME [125]. Activation or polarization of TAMs, from tumor-promoting phenotype (M2) to tumor-suppressing phenotype (M1) (Figure 7), has been found to be a promising immunotherapeutic treatment in oncology [126,127,128].

**Figure 7 pharmaceutics-15-00236-f007:**
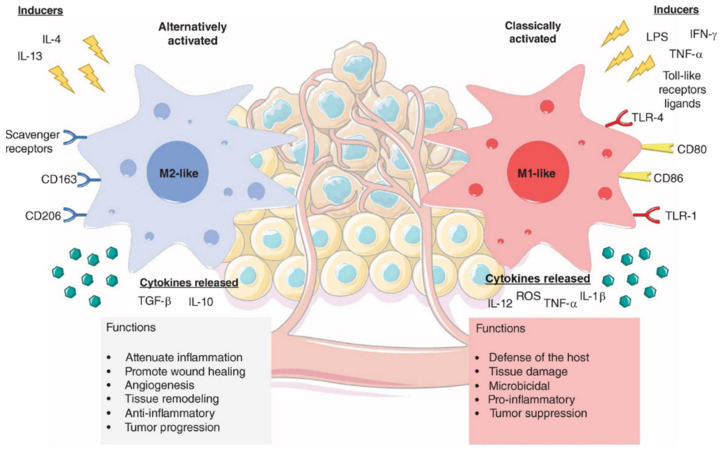
Binary categorization of proinflammatory (M1-like) and anti-inflammatory (M2-like) macrophages in the tumor microenvironment. Reprinted from [126].

As an example, Zanganeh et al. found significant tumor growth inhibition induced by ferumoxytol [129]. This effect was demonstrated to be the result of macrophage polarization into pro-inflammatory M1 phenotypes, along with an enhancement of ROS production leading to increased cancer cell cytotoxicity. A complementary study performed by Zhang et al. found the role of particle charge on the polarizing effect of iron oxide nanoparticles [130]. Both positively charged and negatively charged nanoparticles were able to repolarize macrophages, while neutral nanoparticles did not show significant tumor growth inhibition. The relationship between iron oxide nanoparticle structure and macrophages activation has been investigated by Gu et al., whose work found that higher levels of intracellular iron released from magnetite nanoparticles is the key for superior macrophage polarization and better antitumor performance [131]. Iron oxide nanoparticles with different morphologies have been synthesized for re-polarization therapy, and porous hollow iron oxide nanoparticles were developed by Li et al. and evaluated for TAMs targeting and switching to pro-inflammatory M1-phenotype [132]. Such nanoparticles, combined with 3-methyladenine (3-MA) and modified with mannose for TAMs targeting, were demonstrated to activate the inflammatory factor NF-κB p65 of macrophages responsible for macrophage polarization and subsequent inhibition of tumor growth demonstrated in vivo. Nanoparticles with four different morphologies, namely octapods, cubes, spheres, and plates, were synthesized by Liu et al. and studied regarding inflammasome activation [133]. In their work, they demonstrated that octapods and plates exhibit significantly higher inflammasome-activating capacity compared to cubes and spheres, hence suggesting that particle design must be considered when evaluating new immunotherapeutic formulations based on magnetic nanoparticles.

#### 3.3.2. Ferroptosis

Besides macrophage polarization, iron oxide nanoparticles can exert a cytotoxic effect through a peculiar type of regulated necrosis called ferroptosis. This form of iron-programmed cell death is triggered by high levels of iron ions and ROS, leading to peroxidation of lipids and eventually cell death [134,135,136]. Wen et al., for example, studied the relationship between autophagy pathway and ultrasmall iron oxide nanoparticles-induced ferroptosis [137]. Results showed that these nanoparticles could significantly upregulate the ferroptosis markers in glioblastoma cells, while suggesting that the induced ferroptosis is regulated via Beclin1/ATG5-dependent autophagy pathways. More recently, different-sized iron oxide nanoparticles were investigated by Tian et al. regarding their antitumor effect and toxicity mechanism. Ultrasmall nanoparticles with diameters below 5 nm were found to be more efficient for the generation of ROS in the nucleus due to quicker release of Fe^2+^, though 10 nm nanoparticles displayed the best antitumor effect in vivo. Fernandez-Acosta et al. described the synthesis of SPION functionalized with gallic acid (GA) and polyacrylic acid (PAA) [138], previously used as support for the immobilization of *Trametes versicolor* laccase and subsequent removal of organic dyes [139], and confirmed their intrinsic anticancer activity mediated by ferroptosis execution. Cell death through ferroptosis was confirmed in glioblastoma, neuroblastoma, and fibrosarcoma cells. More recently, Wu et al. [140] found SPION-induced macrophage polarization toward an M1 phenotype and activation of the ferroptosis pathway through the upregulation of P53, a transcription factor responsible for suppressing tumor growth [141].

#### 3.3.3. Chemodynamic Therapy

Chemodynamic therapy (CDT) has recently emerged as a therapeutic strategy taking advantage of the intrinsic sensibility of cancer cells to oxidative damages and the characteristics of the TME to treat cancers with high specificity [142]. The as-mentioned characteristics of the TME include mild acidity, hypoxia, H_2_O_2_ and glutathione (GSH) overexpression, and high nutrient consumption, all of which can modulate the impact of CDT in the context of cancer therapy.

A non-optimized CDT can present a limited therapeutical effect due to the limitations of the TME characteristics. For example, it is well known that H_2_O_2_ is overexpressed in cancer cells (over 100 µM) due to their abnormal growth and metabolism [143]. However, this increased concentration of substrates is still too low for an optimal Fenton’s reaction considering the overexpression of GSH in cancer cells. To overcome this limitation, Huo and coworkers created a subsequent nanoreactor by co-encapsulating glucose oxidase (GOx) and SPION into biodegradable dendritic silica nanoparticles [144]. This work developed nanoparticles that trigger the tumor cell apoptosis by the chemo-Fenton catalysis. In more detail, the liberation of GOx catalyzes glucose conversion in gluconic acid and H_2_O_2_ production (Equation (4)) generates highly toxic hydroxyl radicals by Fenton reaction catalyzed by iron ions.
(4)$$O_{2}+{\;H}_{2}O+\mathrm{glucose}\to \mathrm{gluconic}\;\mathrm{acid}+{\;H}_{2}O_{2}$$

A similar nanocatalyst was developed by the research group of Feng et al. which proposed a therapeutic approach based on composite nanocatalysts made of SPION encapsulating polypyrrole (PPy) and Gox (Fe_3_O_4_@PPy@Gox) [145]. This approach successfully produced chemodynamically triggered cancer hyperthermia, dual-modality diagnostic imaging-guided and nanocatalytic therapy based on the chemo-Fenton catalytic reaction (as mentioned above). Similarly, Dong and colleagues created a magnetic targeting nanoplatform by encapsulating ultrasmall γ-Fe_2_O_3_ nanoparticles and GOx to dendritic mesoporous silica (DMSN) spheres (γ-Fe_2_O_3_-GOx-DMSN) [146].

Another CDT limiting factor relies on the ability of GSH to quench oxidative molecules. To circumvent this limitation, Zhao et al. proposed a ROS-activatable liposome loaded with hexadecyl-oxaliplatin carboxylic acid (HOC) and Fe_3_O_4_ nanoparticles (RALP@HOC@Fe_3_O_4_) [147]. In this model, HOC acts as a prodrug that would be metabolized by GSH into oxaliplatin (OXA) while inducing the starvation of GSH in the cancerous cells. This breakdown of the tumor cells weakening the redox equilibrium induces an increased generation of H_2_O_2_ while the released SPION provide a substrate for the Fenton’s reaction. This model has been tested on a CT26 xenograft murine model and showed a significant reduction in tumor growth speed. A similar strategy was proposed by Chen et al. This study saw the development of an Fe_3_O_4_ nanoplatform coated by human serum albumin (HSA) and β-lapachone (Lapa) [148]. This platform (Fe_3_O_4_-HSA@Lapa NP) overexpressed NQO1 enzyme and catalyzed the amount of released Lapa, which resulted in a large amount of O^2•^, which was then converted into H_2_O_2_ by superoxide dismutase (SOD). Conversely, in the acidic endolysosome environment, ferrous ions from Fe_3_O_4_ combined with overproduced H_2_O_2_ catalyze the formation of OH^•^ by the Fenton reaction, which finally led to cell death. Additionally, the use of NADPH in the futile Lapa cycle led to GSH depletion, altering the antioxidant defense system and increasing the sensitivity of cancer cells to OH^•^ produced in the Fenton chemical reaction, hence intensifying the anticancer impact both in A549 cells and the xenografted in vivo mouse model A459.

The Fenton reaction appears to be pH-dependent (optimal pH around 3) [149]; the higher pH value of TME (pH 5–7) is thus not suitable for this catalytic conversion. To counteract this barrier, Shi et al. created an acid-unlocked nanoplatform loaded with the pH-sensitive drug tamoxifen (TAM) (FePt@FeOx@TAM-PEG) [150]. The above work showed the release of TAM drug for the inhibition of mitochondrial complex I, which after several cascade events causes the increase in intracellular H^+^. Due to the decrease of pH, FePt@FeOx nanocatalyst may also be released, which would cause a Fenton-like reaction. An increase in DNA damage has been observed in cancer cells treated with FePt@FeOx@TAM-PEG. In addition, in vivo treatment on 4T1 xenografted mice has been successfully carried out using the increased ROS production. Another strategy involving the modulation of the TME’s pH has been described by Liu et al. In this study, SPION based RNAi modulated the glycolysis pathway by the monocarboxylate transporter four protein (MCT4) silencing (Figure 8) [151]. Overexpression of MCT in cancer cells induces an increase of lactate/H+ efflux process and thus stabilization of intracellular pH. Consequently, targeting the MCT protein family appears as a promising strategy to decrease the intracellular pH and promote the Fenton and Fenton-like reaction inside the cancer cells.

The high metabolic activity of cancer cells in conjunction with the limited vascularization of this area can, in some cases, induce hypoxia, thus altering the presence of ROS and consequently decreasing the CDT impact. Recently, Li et al. described a SPION platform enriched with manganese ions and pyropheophorbide (PPA) [152] and providing high glutathione-peroxidase-like, Fenton-like, and catalase-like activity. The rationality between this combination of functions is that catalase-like activity induces a catalyzed conversion of H_2_O_2_ to O_2_ and subsequently decreases the hypoxic state of TME. In addition, the glutathione-peroxidase-like activity was involved in the reduction of GSH concentration in the TME and subsequently the resistance of cancer cells to oxidative damage. Finally, Fenton-like activity gives the platform the ability to catalyze H_2_O_2_ conversion in OH^•^ and exert a CDT activity.

Additional studies focused on the relevance of electric-assisted Fenton reactions. As an example, Chen’s team described SPION decorated with platinum nanocrystals (Fe_3_O_4_@Pt NP), which combine CDT with electrodynamic therapy (EDT) [153]. This platform showed an ability to produce ROS due to an electric-field-induced catalysis on the surface of platinum. In the meantime, Fe^2+^/Fe^3+^ from Fe_3_O_4_@Pt NP may stimulate the Fenton reaction to turn intracellular H_2_O_2_ into ROS and consume GSH (inhibiting ROS clearance). As a result, a significant tumor inhibition capability was seen both in vitro and in vivo when exposed to an electric field. Another strategy involving the extra-production of ROS by platinum has been described by Ma et al., who presented multifunctional Janus nanoparticles (Fe_3_O_4_-Pd JNPs) that combine ultrasmall Fe_3_O_4_ and Pd nanosheets for the treatment of breast cancer [154]. In this report, an increased production of ROS was found due to the catalytic reaction of Fe_3_O_4_ nanoparticles and Pd nanosheets in the acidic TME.

**Figure 8 pharmaceutics-15-00236-f008:**
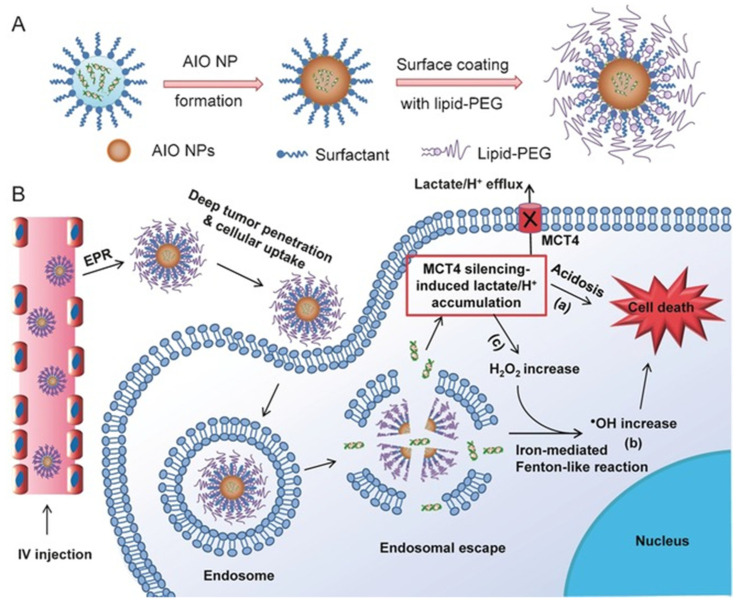
Illustration of (**A**) SPION-based RNAi platforms (AIO: amorphous iron oxide) and (**B**) their interaction with tumor cells. Upon i.v. administration, nanoparticles accumulate in tumor cells through the EPR effect. Internalization of the nanoparticles within the endosome and subsequent release of iron ions leads to osmotic pressure and/or endosomal membrane oxidation. The resulting endosomal escape induces the release of RNAi and iron ions, resulting in MCT4 silencing and oxidative stress via the Fenton-like reaction. Reprinted from [151].

#### 3.3.4. Light-Mediated Therapy

Besides taking direct advantage of the TME to trigger Fenton reaction and iron-catalyzed reactions leading to the formation of ROS, combining SPION with photosensitizers capable of inducing the release of ROS or heat upon exposure to UV-visible or near infra-red (NIR) light is a growing research topic in the field of cancer therapy. Illumination of a photosensitizer using UV light can trigger photochemical reactions with oxygen leading to the production of ROS (Figure 9a), responsible for cancer cell death. Promoting ROS production using this mechanism is generally referred to as photodynamic therapy (PDT). Alternatively, photosensitizers can release heat when irradiated with NIR light and cause the destruction of cancer cells through several mechanisms (Figure 9b). This strategy is known as photothermal therapy (PTT).

**Figure 9 pharmaceutics-15-00236-f009:**
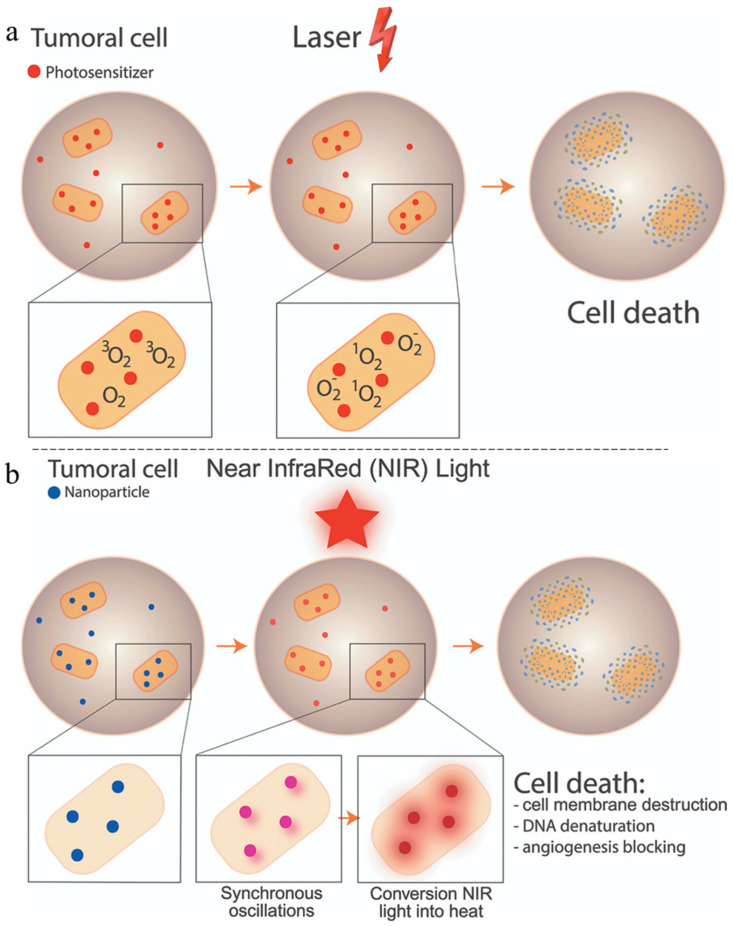
Illustration of (**a**) photodynamic therapy mechanism and (**b**) photothermal therapy mechanism. Reprinted from [155].

The principle of PDT is the formation of ROS by irradiation of photosensitizers with a specific wavelength of light energy, usually a laser. By absorbing a photon, the photosensitizer reaches an excited singlet state that rapidly transits to a triplet state through an inter-crossing system. The triplet state has a longer half-life, allowing interaction with ground state dioxygen or other cellular compounds [156,157]. There are two types of photodynamic processes, the first, called type I, is the transfer of an electron from the triplet state photosensitizer to a cellular compound such as a protein, lipid, or nucleotide resulting in the formation of ROS species. The second type, type II, is the energy transfer from the triplet state to a dioxygen molecule in the ground state, leading to the formation of cytotoxic singlet oxygen [158].

In the best-case scenario, a photosensitizer would generate singlet oxygen when exposed to a specific wavelength, while being harmless under normal conditions, and would be able to precisely target cancer cells due to certain characteristics. So far, three generations of PS have been developed. The first generation consists of a natural mixture of hematoporphyrin derivatives. The second generation is pure synthetic compounds that absorb light in the vis-NIR. The third generation refers to photosensitizers with additional features that enhance their biomedical uses such as antibodies or nanotechnology [159].

PDT is a minimally invasive technique which, as for other techniques implying light emission, is limited by light penetration through tissues. Employing photosensitizers absorbing within the therapeutic window (i.e., range of wavelengths where light has maximum penetration in tissues [160]) is mandatory. Common photosensitizers are excited using wavelengths between 650 and 800 nm, which stands within the window. However, PDT remains limited to treating skin cancers or tumors located on a mucous membrane accessible to an endoscope [161]. A way around this restriction is the use of optical fibers to reach most of the body cavities, even for vertebral metastasis as described by Fisher et al. by combining PDT with vertebral cement augmentation [162]. Besides optical properties, PDT outcome is dependent on the nature of the photosensitizers, which are generally known for their hydrophobicity, resulting in poor circulation half-life and restrained therapeutic efficacy. Furthermore, photosensitizing molecules lack specificity for targeting tumor cells. This could lead to mid- and long-term adverse effects, such as whole-body photosensitivity. The use of SPION as nanocarriers is a way to overcome the targeting and hydrophobicity issues [156]. SPION are mainly used as nanocarriers due to their ability to penetrate and accumulate into cancer cells by EPR effect. Specific targeting is also possible with adapted coating, such as transferrin, antibodies, aptamers, hyaluronic acid, folate, and targeting peptides [163]. As an example, Li et al. described the use of SPION synthesized by thermal decomposition in polyol solvents as nanocarriers for doxorubicin and chlorin e6 (Ce6). Authors found an enhanced cell uptake due to the formation of clusters loaded with Ce6 through physical adsorption and coordination with surface atoms. In vivo PDT efficacy was evaluated in mouse subcutaneous melanoma grafts, and enhanced PDT efficacy was demonstrated to be the result of Dox specific release in cancer cells due to the acid-sensitive hydrazone linkage between the drug and the nanocarrier [164].

PTT is a selective and minimally invasive therapeutic strategy used for cancer cell thermal ablation. Thermal ablation consists of a local increase in temperature above 42 °C, for short time periods. Depending on the temperature threshold, cancerous cells can be killed over long periods of time (at 42 °C) or within shorter time windows (i.e., 4–6 min) when increasing the temperature over 50 °C [165].

Heat production comes from the irradiation of photothermal agents, which dissipate the received energy in the form of heat by various processes (Figure 8B). As an example, metallic nanoparticle-induced heat generation comes from synchronized oscillations of the conduction band electrons generated by the irradiation (LSPR). There is a large panel of photothermal agents (PAs) of different nature, and their ability to increase temperature depends on their photothermal conversion efficiency. Similar to PDT, the excitation of PAs is performed using specific wavelengths within the therapeutic window. Although PAs are commonly excited using longer wavelengths (compared to PDT), either in the NIR light range (from 650 to 1024 nm), although most use irradiation wavelengths of 808 nm and 980 nm. The advantages of PTT are its low invasiveness, spatial accuracy, and the fact that it does not require the presence of intracellular O_2_ unlike PDT. Despite this, this technique suffers from some constraints, the most obvious being the shallow penetration depth of the NIR light into the tissue. Even though there are NIR-I (750–1000 nm) and NIR-II (1000–1700 nm) windows, the penetration depth remains in the order of the cm [165,166,167].

SPION can be used in two ways in PTT. Similar to PDT, they can serve as nanocarriers for other PAs by providing stability and a way to enter and accumulate in cancer cells via the EPR effect. Unlike in PDT, SPION can directly serve as therapeutic agents, as they generate heat under NIR irradiation [168]. The applicability of PEG-starch-SPION for PTT was studied by Amatya et al. for the treatment of glioblastoma [169]. In this study, PEGylation was employed to extend the plasma half-life of SPION, which further led to an increase in tumor accumulation. Laser irradiation of the tumor site using a 885 nm wavelength 4 h after the administration of SPION led to a significant temperature increase (above 45 °C) which was sufficient for effective inhibition of tumor growth. Dheyab et al. developed gold-coated SPION, generating heat via localized plasmon resonance, and evaluated cell viability following photothermal therapy treatment [170]. MCF-7 cells treated with Au-SPION, and irradiated with NIR light (808 nm, 200 mW, 10 min) displayed a significant decrease in viability due to the PTT treatment, demonstrating that such systems are promising for image-guided PTT treatment of breast cancer. Combining PTT with PDT is also possible through the coating of SPION with photosensitizers generating ROS under irradiation. Zhang et al. showed the use of green and biocompatible components to develop glucose oxidase (GOx) and polydopamine (PDA) coated SPION. GOx is a photosensitizer which is able to generate high levels of H_2_O_2_ that can react with SPION to form OH^●^ radicals via a Fenton-like reaction. A PTT-ROS dual-modality system was hence constructed, in which PDA was used as both cross-linking platform and PA, and SPION were used as carriers and catalyzers for the generation of radicals. In vivo study on 4T1 tumor-bearing Balb/c mice was performed and confirmed the efficient combination between ROS generation and PTT, characterized by selective anticancer effect and significant tumor suppression. Another way of improving the selectivity of PA-bearing SPION towards the tumor site is the use of magnetically guided targeting approach. This strategy was employed by Wang et al.who engineered SPION coated with silicone dioxide and indocyanine green (ICG) as PAs. Particle accumulation in cancer cells was enhanced through the combination of magnetic guiding and the natural ability of neutrophils to infiltrate tumor tissues [171]. Due to these combined strategies, the authors found stronger contrast enhancement in the tumor region, as well as a marked enhancement of the curative effect after the photothermal process.

#### 3.3.5. Combination with Radiation Therapy

Nowadays, radiation therapy, more commonly called radiotherapy (RT) appears as a major strategy against cancer [172,173,174]. The importance of RT in the therapeutic handling of cancer is such that it is estimated that more than 50% of the patients treated for cancer will require at least one dose of RT during their treatments [175,176]. The mechanism of action of radiotherapy is based on irradiation and consequently the induction of direct or indirect damage induction to many cellular structures such as DNA [177], proteins [178], lipids [179,180], or the water constituting the cytoplasm [181,182]. 

Recently, numerous advances (e.g., FLASH RT, proton therapy, fractionation of the treatment, intensity modulated RT) have been achieved in the field of RT to increase accuracy, specificity, and efficacy in radiation treatment [183,184,185,186,187]. However, the main limitation of RT is still the potential lack of precision during the irradiation of the tumors, the radio-resistance developed in some cases by the tumors, and the radiosensitivity surrounding normal tissues. Thus, the use of radiosensitizers (e.g., molecules or particles) appears as a promising approach to enhance RT efficiency, thus reducing the normal tissue damage while keeping the same biological effect against the cancerous cells.

Owing to a higher attenuation coefficient, the radiosensitization strategies have mainly focused on high atomic number nanoparticles (High Z NPs) [188,189,190,191,192,193,194,195,196]. However, some emerging studies used nanomaterials that have the ability to improve RT efficiency through other mechanisms such as SPION. One of the main mechanistic descriptions of the radiosensitization process related to SPION appears to be the Fenton’s and Haber–Weiss’ reactions. Under physiological conditions, most of the intracellular iron appears to be bound to iron binding proteins such as ferritin, resulting in only trace amounts of free intracellular iron. However, under the presence of SPION, several works mention that the internalization of iron oxide nanoparticles can lead to an increase in the concentration of free intracellular iron which may be involved in the production of excessive ROS that could further alter the cellular homeostasis [197]. This extra production of ROS could finally lead to irreversible cellular injury or to cell apoptosis, especially when SPION are combined with additional treatment such as RT [198,199]. In addition to Fenton’s and Haber–Weiss’ reactions related considerations, several mechanisms have been proposed to explain the radiosensitizing potential of SPION through the last years. 

Shetake et al. reported that oleic acid modified SPION could interact with HSP60 and HSP90 which are key proteins in the mitotic process as well as in the DNA repair process [200]. This observation suggests SPION related radiosensitization mechanism involving the direct interaction between a SPION and a protein. He et al. suggested that SPION can interfere with the electron transport chain within mitochondria (METC) of cancerous cells, which also affects ATP synthesis, the mitochondrial membrane potential, calcium microdistribution, and favor the induction cell death process. In addition, due to the reduced antioxidative ability of cancer cells, SPION are also suggested to cause the generation of reactive oxygen species in mitochondria which are all elements that can amplify the effect of RT [201].

Several authors investigated the radiosensitizing properties of SPION functionalized with chemotherapeutic components as nanocarriers [202,203,204,205]. As an example, Popescu et al. studied the radiosensitization potential of doxorubicin-loaded polyethylene glycol-encapsulated SPION on HeLa cells after 6 MV X-rays irradiation [206]. This study showed a drastic decrease in clonogenic survival in the case of doxorubicin-loaded SPION compared to doxorubicin-free SPION followed by 6 MV X-rays beams explained by the synergetic effect of irradiation and the intracellular release of doxorubicin causing additional DNA damage. A similar study has been conducted by Kang et al. The authors developed a SPION platform functionalized with folate to target cancer cells and paclitaxel as a chemotherapeutic agent. The developed platform has shown a radiosensitization potential for 40 MeV proton beam exposure [207].

Most of the reports that have been currently presented focused on the use of SPION with an external radiation beam. However, Stanković et al. synthesized and evaluated ^131^I-labeling APTES functionalized SPION and took advantage of the radionuclide radiopharmaceuticals for nanoparticle mediated brachytherapy. This report investigated the potential of this particular platform with regard to the brachytherapy aspect but also showed that this effect could be enhanced when applied in combination with magnetic induced hyperthermia. Nevertheless, they demonstrated the use of ^131^I-APTES functionalized SPION without magnetic-induced hyperthermia was sufficient to induce a significant lowering of the tumor growth rate [208]. Similarly, Zuk et al. explored the potential of SPION containing a radioactive isotope of gold and functionalized with trastuzumab in order to target HER-2 positive cancer cells [209]. This strategy allowed the authors to induce a significant regression of spheroid growth. The synergy between MH and RT was also demonstrated by the group of Lin that synthesized and evaluated a magnetic silica encapsulated multi-core SPION platform coated with folic acid and L-selenocystine [210]. This platform, tested on Hela cells, human breast cancer cells (MDA-MB-231), and human umbilical vein endothelial cells (HUVEC) showed a high saturation magnetization and an ability to produce increased ROS amount when exposed to X-rays beams. Taken together, these features showed a synergistic effect on the elimination of cancer cells and highlighted the therapeutic interest to combine MH and RT.

The diversity of results regarding the radiosensitization potential of SPION among the multiple research groups presented in this review highlights the importance of the exposure parameters used for the radiosensitization (including the type of irradiation, the cell line used, the physico-chemical properties of SPION (size, shape, and coating), and the conditions tested (NPs, concentrations and incubation times). Such a wide range of parameters underscores the need to optimize and determine an optimal formulation of SPION for radiosensitization purposes. It should be noted that this search for an optimal formulation is made even more difficult by the diversity of the elements included in the publications of the different groups. At present, to our knowledge, there have been no large-scale studies evaluating and comparing the radiosensitization potential of a large number of formulations on a multitude of models. 

## 4. Conclusions

In recent years, considerable progress has been made in the synthesis of iron oxide nanoparticles for biomedical applications and more specifically in cancer therapy fields. Various original methods have been developed to control their size, shape, and composition, all these features influencing their final magnetic and biological behavior. In addition, our growing ability to control their surface chemistry through different ingenious strategies has extended their scope of action by allowing, for example, the conjugation of drugs (covalent or non-covalent; on the surface or within self-assembled systems), or to stimulate given biochemical process inducing local toxicity.

Despite all the above mentioned advances and the increasing degree of sophistication of the proposed systems, it is clear that the integration of multifunctional SPION into the clinical landscape is still missing. Based on this observation, one may suppose that an extensive comprehension of the events involved in the transport of the nanocarrier to the targeting site appears mandatory. For example, in the context of tumoral treatment, the number of particles that actually reaches the tumor, the control of their elimination pathways, their biological properties (including their toxicity, biocompatibility, and interaction with plasma proteins) are crucial factors to be considered when studying a SPION formulation. To this, we can add the uncertainty related to the use of ectopic murine tumor models (more particularly in the case of EPR effect), thus implying a reflection on the relevance of some preclinical strategies.

Nevertheless, because of their properties and their intrinsic imaging capabilities (including MPI), SPION appear as particularly suitable for theranostic applications, and the studies gathered in this review propose exciting new perspectives in the field of SPION-based cancer treatment. Among these novel applications, the generation of ROS from nanoparticles, either alone or in combination with other therapeutic modalities, is an emerging feature gaining considerable interest over the past years due to the particular role of ROS in cancer progression. SPION-induced production of ROS through the Fenton/Haber–Weiss reaction hence appears as a prominent asset to guide future directions in the field of SPION-based therapies. Numerous theranostic SPION-based platforms have been designed and have shown promising results, alone or in combination with other therapeutic strategies (RT, chemotherapy, …). While some strategies appear to be hampered by the limited dose that actually reaches the tumor, original strategies were developed to increase the local particle concentration (sonoporation, for example). Overall, considering all the possibilities offered by SPION, these materials should play a key role in the field of cancer therapy in the near future.

## Figures and Tables

**Figure 1 pharmaceutics-15-00236-f001:**
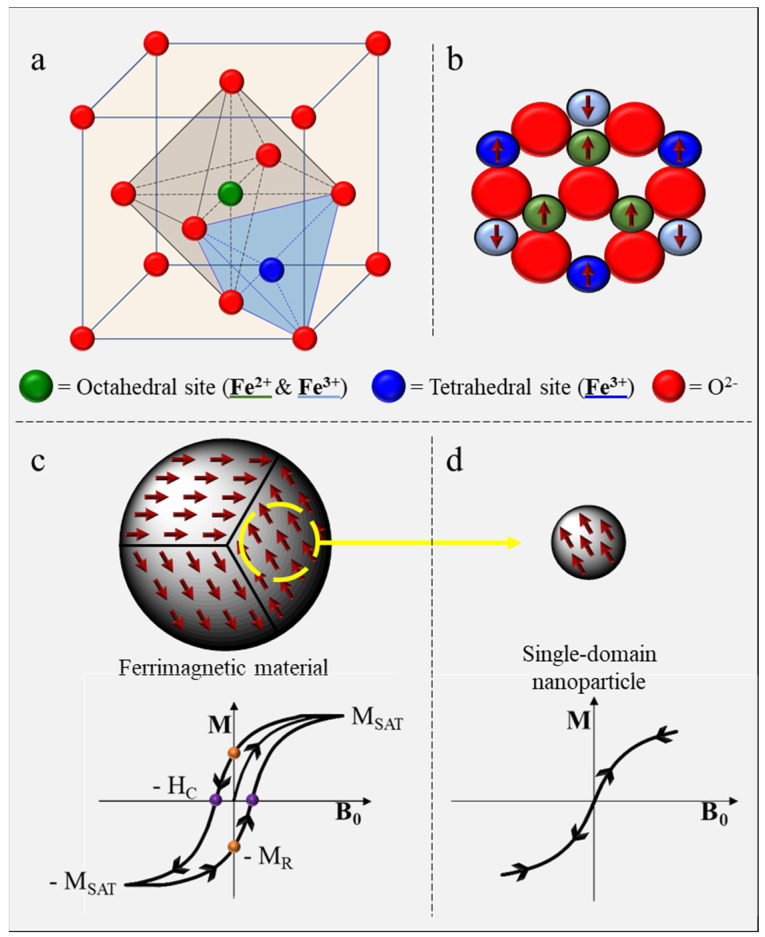
(**a**) Illustration of the face-centered cubic arrangement of O^2−^ anions and location of ferric and ferrous ions in octahedral and tetrahedral sites; (**b**) illustration of the ferrimagnetic network formed by ferrous and ferric ions in magnetite; (**c**) illustration of domain formation in ferrimagnetic material and resulting magnetization behavior; (**d**) illustration of a single-domain nanoparticle and resulting superparamagnetic curve.

**Figure 2 pharmaceutics-15-00236-f002:**
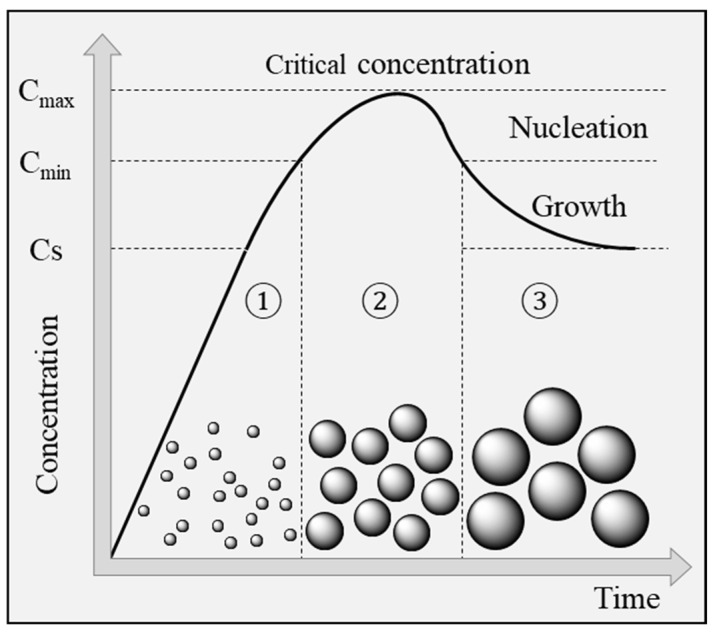
Lamer diagram schematic showing an increase in monomer concentration (Step 1), the nucleation (Step 2), and growth phenomenon (Step 3) in function of monomer concentration and time.

**Figure 4 pharmaceutics-15-00236-f004:**
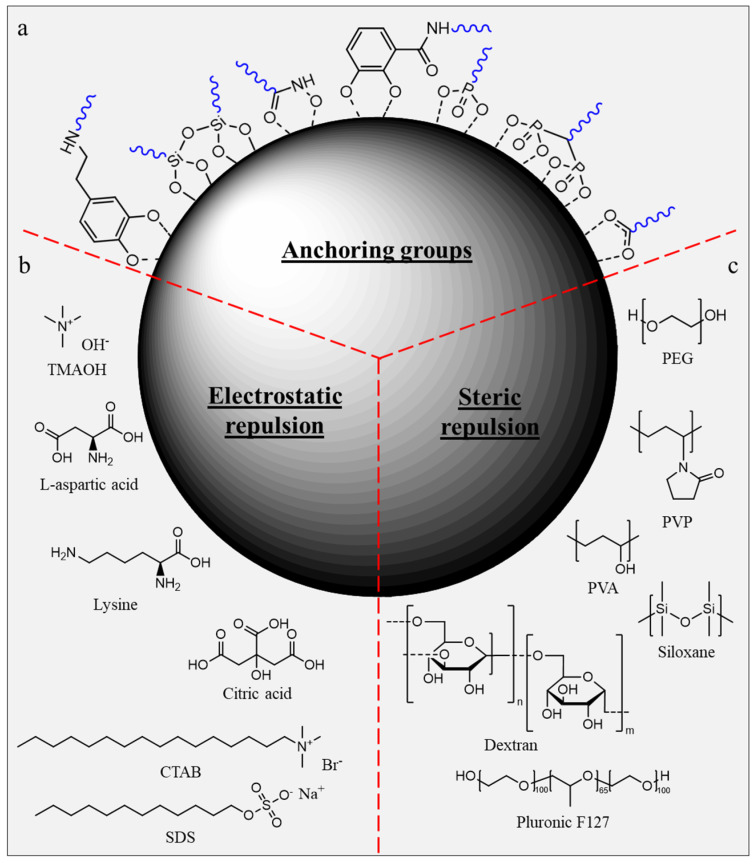
(**a**) Anchoring groups grafted on an iron oxide surface, from left to right, dopamine, siloxane, hydroxyamate, 2,3-dihydroxybenzamide, mono- and bis-phosphonate, and carboxylate; (**b**) examples of capping agents providing electrostatic stabilization; (**c**) polymer coatings providing steric stabilization.

## Data Availability

Not applicable.

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
