# Peer review of "Superparamagnetic Iron Oxide Nanoparticles (SPION): From Fundamentals to State-of-the-Art Innovative Applications for Cancer Therapy"

_pharmaceutics, 2023, doi:10.3390/pharmaceutics15010236_

Round 1
Reviewer 1 Report
In this manuscript the authors reported the review of comprehensive description of the fundamental aspects of SPION formulation and highlights various recent approaches regarding in vivo applications in the field of cancer therapy. Some interesting results are obtained. I therefore recommend an acceptance for publishing after minor style revisions.
Author Response
Authors would like to thank the Reviewer 1 for his/her constructive remarks on this review article, the comments were highly appreciated, and the authors hope to meet their expectations after the revision.
Reviewer 2 Report
This is a well written paper illustrating precisely the actual state and applications of iron oxide nanoparticles. My feeling is that the directed transport of drug-magnetic nanoparticle conjugate in the presence of constant magnetic field (e.g. paper Biomacromolecules 2013, 14, 828–833) could be better stressed.
Other minor points:
Fig 2 (first figure): The figure number should be 1. Also, Figure 2d: it could be good to indicate with arrows that the forward and backward curves overlap.
Fig 2: 1, 2 and 3 are not addressed in the figure legend.
Table 1, Electrochemical synthesis: 'Controllable particle size'
and 'Inability to reproduce' are somewhat contradictory
Line 346-348: wrong sequence of OH- and OH and wrong
description of reaction (3).
Figure 8 label raises questions.
Some editorial imperfections can be found, e.g. missing words and characters.
Author Response
Authors would like to thank the Reviewer 2 for his/her constructive remarks on this review article, the comments were highly appreciated, and the authors hope to meet their expectations after the revision.
This is a well written paper illustrating precisely the actual state and applications of iron oxide nanoparticles. My feeling is that the directed transport of drug-magnetic nanoparticle conjugate in the presence of constant magnetic field (e.g. paper Biomacromolecules 2013, 14, 828–833) could be better stressed.
Other minor points:
Fig 2 (first figure): The figure number should be 1. Also, Figure 2d: it could be good to indicate with arrows that the forward and backward curves overlap.
Correction was performed concerning the figure number. Arrows were added on both figures 2c and 2d to indicate the curve behaviors.
Fig 2: 1, 2 and 3 are not addressed in the figure legend.
The legend of the figure was modified to describe the three steps of particle formation.
Table 1, Electrochemical synthesis: 'Controllable particle size' and 'Inability to reproduce' are somewhat contradictory
As suggested by reviewer 3, table 1 was removed from the manuscript and the reference from the figure legend was inserted within the text (ref. 23)
Line 346-348: wrong sequence of OH- and OH and wrong description of reaction (3).
Correction was made in the text and in equation 3.
Figure 8 label raises questions.
A description of the global scheme was inserted in the figure label.
Some editorial imperfections can be found, e.g. missing words and characters.
Corrections were made in the manuscript regarding editorial imperfections.
Reviewer 3 Report
The manuscript “Superparamagnetic iron oxide nanoparticles (SPION): From fundamentals to state-of-the-art innovative applications for cancer therapy” highlights various recent approaches to in vitro and in vivo applications of superparamagnetic iron oxide nanoparticles (SPION) in the field of cancer therapy. The review includes sections that briefly describe the fundamentals of superparamagnetism, methods for the synthesis and modification of the surface of magnetic nanoparticles, and the mechanism of protein corona formation in biological media. The review contains all the main areas of use of SPION in cancer therapy (hyperthermia, drug delivery, PDT, PTT, etc.). Unfortunately, the text contains many phrases of the same type (sometimes repeated verbatim) that are present in the review previously published by the authors [23] Expert Opin. Drug Deliv. 2022, 19, 321–335 (doi:10.1080/17425247.2022.2047020), which reduces the novelty of this manuscript. A positive feature of the work is the mention of a fairly new direction in therapy associated with the use of low-frequency alternating magnetic fields (magneto-mechanical therapy). Considerable emphasis is placed on describing the mechanisms of reactive oxygen species (ROS) production (Haber-Weiss and Fenton reactions) and ROS-mediated therapies. However, the presentation of the material does not seem to be sufficiently consistent and clear. Some issues should be considered that are presented below.
1. I propose to remove or replace a number of fragments of the text that repeat the material published by the authors in the review [23] Expert Opin. drug deliv. 2022, 19, 321–335 (doi:10.1080/17425247.2022.2047020) (these are either the same type of phrases, or the same type of description of examples of obtaining and researching materials), for example, lines 48-50, 60-64, 371-375, 377-381 , 393-399, 402-410, 562-569, ... Table 1 should be either substantially supplemented with new material, or removed and simply referred to the previous review [23].
2. Sections ‘3.1. Drug delivery' and '3.2. Magnetically-activated therapy’ seem to be quite close in their content. Both sections present similar examples of obtaining materials based on magnetic particles (MNPs), the essence of which is due to their specific behavior in alternating magnetic fields (AMF) (movement or heating). Section 3.2 describes the application of low frequency AMFs and high frequency AMFs (magnetic hyperthermia (MH)), followed by examples of drug loaded materials (line 552-569) that logically should have been discussed in section 3.1. However, Section 3.1 discusses drug-loaded materials with drug release are stimulated by AMF. In my opinion, sections 3.1 and 3.2 should be combined into a single logical block, and the examples of the materials described in them should be given in sequence.
3. Section ‘3.5 Combination with radiation therapy’ contains data on magnetic materials whose toxic effect arise from ROS products. Thus, this section can be considered as a subsection of section “3.3. ROS-mediated therapies”. In addition, the example from section 3.2 (lines 570-577) seems should be discussed in the section ‘Combination with radiation therapy’.
4. Since much attention is paid to the description of the mechanisms of ROS production and ROS-mediated therapies in this review, in conclusion, in order to highlight the importance of this direction, a discussion on this topic should be given.
5. There are inaccuracies in the text that should be corrected (for example, line 507). The phrase: “an alternating magnetic field (AMF)” appears many times in the text. It is enough to give the transcript once, and then use only the abbreviation.
The manuscript could be accepted for publication in Pharmaceutics after major revision.
Author Response
Authors would like to thank the Reviewer 3 for his/her constructive remarks on this review article, the comments were highly appreciated, and the authors hope to meet their expectations after the revision.
The manuscript “Superparamagnetic iron oxide nanoparticles (SPION): From fundamentals to state-of-the-art innovative applications for cancer therapy” highlights various recent approaches to in vitro and in vivo applications of superparamagnetic iron oxide nanoparticles (SPION) in the field of cancer therapy. The review includes sections that briefly describe the fundamentals of superparamagnetism, methods for the synthesis and modification of the surface of magnetic nanoparticles, and the mechanism of protein corona formation in biological media. The review contains all the main areas of use of SPION in cancer therapy (hyperthermia, drug delivery, PDT, PTT, etc.). Unfortunately, the text contains many phrases of the same type (sometimes repeated verbatim) that are present in the review previously published by the authors [23] Expert Opin. Drug Deliv. 2022, 19, 321–335 (doi:10.1080/17425247.2022.2047020), which reduces the novelty of this manuscript. A positive feature of the work is the mention of a fairly new direction in therapy associated with the use of low-frequency alternating magnetic fields (magneto-mechanical therapy). Considerable emphasis is placed on describing the mechanisms of reactive oxygen species (ROS) production (Haber-Weiss and Fenton reactions) and ROS-mediated therapies. However, the presentation of the material does not seem to be sufficiently consistent and clear. Some issues should be considered that are presented below.
I propose to remove or replace a number of fragments of the text that repeat the material published by the authors in the review [23] Expert Opin. drug deliv. 2022, 19, 321–335 (doi:10.1080/17425247.2022.2047020) (these are either the same type of phrases, or the same type of description of examples of obtaining and researching materials), for example, lines 48-50, 60-64, 371-375, 377-381 , 393-399, 402-410, 562-569, ... Table 1 should be either substantially supplemented with new material or removed and simply referred to the previous review [23].
Based on the advice of Reviewer 3, the concerned lines have been rewritten and additional examples have been added within the manuscript regarding the drug delivery section. In addition, table 1 was removed from the manuscript and referred from the previous review.
Sections ‘3.1. Drug delivery' and '3.2. Magnetically-activated therapy’ seem to be quite close in their content. Both sections present similar examples of obtaining materials based on magnetic particles (MNPs), the essence of which is due to their specific behavior in alternating magnetic fields (AMF) (movement or heating). Section 3.2 describes the application of low frequency AMFs and high frequency AMFs (magnetic hyperthermia (MH)), followed by examples of drug loaded materials (line 552-569) that logically should have been discussed in section 3.1. However, Section 3.1 discusses drug-loaded materials with drug release are stimulated by AMF. In my opinion, sections 3.1 and 3.2 should be combined into a single logical block, and the examples of the materials described in them should be given in sequence.
We agree about the similarity of contents described in these two sections. However, we differentiated the two sections based on the nature of the magnetic field used for the different therapies, i.e. constant magnetic field applied in the case of magnetically assisted drug delivery (in section 3.1) compared to alternating magnetic field (at either low or high frequencies) used for magnetically-activated therapies (in section 3.2). A paragraph was transferred from section 3.1 to 3.2 (paper from Chen et al. ref 121 in the corrected manuscript) as it mentions AMF actuation.
Section ‘3.5 Combination with radiation therapy’ contains data on magnetic materials whose toxic effect arise from ROS products. Thus, this section can be considered as a subsection of section “3.3. ROS-mediated therapies”. In addition, the example from section 3.2 (lines 570-577) seems should be discussed in the section ‘Combination with radiation therapy’.
We agree that the section “combination with radiation therapy” is a subsection from the “ROS-mediated therapies”, change was made in the manuscript. The example from Lin et al. was also moved from section 3.2 to section 3.3.5 (lines 899-906) in the corrected manuscript).
- Since much attention is paid to the description of the mechanisms of ROS production and ROS-mediated therapies in this review, in conclusion, in order to highlight the importance of this direction, a discussion on this topic should be given.
An additional comment was added within the conclusion to highlight the potential of ROS-mediated therapies in future studies using SPION (Lines 942-947).
- There are inaccuracies in the text that should be corrected (for example, line 507). The phrase: “an alternating magnetic field (AMF)” appears many times in the text. It is enough to give the transcript once, and then use only the abbreviation.
Changes were made in the text concerning the transcript of “alternating magnetic field” and other transcripts within the manuscript.
Round 2
Reviewer 3 Report
The authors have made acceptable changes. There are some typos in the text (for example, line 543: a colon should probably be put after “These issues include”).
After technical revision, the manuscript may be published in Pharmaceutics.